# Understanding Alzheimer disease's structural connectivity through explainable AI

**Achraf Essemlali**[*]             Achraf.Essemlali@USherbrooke.ca
**Etienne St-Onge**[*]              Etienne.St-Onge@USherbrooke.ca
**Maxime Descoteaux**[†]           Maxime.Descoteaux@UShrebrooke.ca
**Pierre-Marc Jodoin**[†]          Pierre-Marc.Jodoin@USherbrooke.ca
*Computer science department, University of Sherbrooke, Sherbrooke, Canada*

## Abstract

In the following work, we use a modified version of deep BrainNet convolutional neural network (CNN) trained on the diffusion weighted MRI (DW-MRI) tractography connectomes of patients with Alzheimer's Disease (AD) and Mild Cognitive Impairment (MCI) to better understand the structural connectomics of that disease. We show that with a relatively simple connectomic BrainNetCNN used to classify brain images and explainable AI techniques, one can underline brain regions and their connectivity involved in AD. Results reveal that the connected regions with high structural differences between groups are those also reported in previous AD literature. Our findings support that deep learning over structural connectomes is a powerful tool to leverage the complex structure within connectomes derived from diffusion MRI tractography. To our knowledge, our contribution is the first explainable AI work applied to structural analysis of a degenerative disease.

**Keywords:** Structural connectome, diffusion weighted MRI, deep learning, saliency maps, Alzheimer's Disease

## 1. Introduction

Early detection of neurodegenerative diseases like Alzheimer's Disease (AD) along with proper treatments can delay its progression (Livingston et al., 2017; Weller and Budson, 2018). Several approaches have been explored to better predict, detect and understand the disease. These approaches include biological markers (Hampel et al., 2008; Kapaki et al., 2007; Irizarry, 2004; Blennow et al., 2015; Patel et al., 2011; Zetterberg, 2008; Mattsson et al., 2009; Gomar et al., 2011; Gomez-Isla and Frosch, 2019), blood-based bio-markers (Henriksen et al., 2014; Thambisetty and Lovestone, 2010; Mayeux and Schupf, 2011; Doecke et al., 2012), neuro-psychological tests (McKhann et al., 1984; Tierney et al., 2005; Jacobs et al., 1995; Weintraub et al., 2012), artificial intelligence algorithms on medical images (Liu et al., 2014; Moradi et al., 2015; Li et al., 2019; Jo et al., 2019; Lee et al., 2019; Litjens et al., 2017; Liu et al., 2018). Besides, magnetic resonance imaging (MRI) has been a modality of choice for AD diagnostics and has demonstrated its significance (Vemuri and Jack, 2010). Most MRI-based techniques for studying AD can be grouped under two main categories : i) MRI anatomical images analysis and ii) structural and functional connectomes

---

[*] Contributed equally
[†] Contributed equally

(connectivity matrices) (Contreras et al., 2015). Standard techniques employing MRI can be distinguished between: clinical analysis (Engelborghs, 2013; Cummings et al., 2019), segmentation techniques (Biju et al., 2017) and machine/deep learning algorithms (Jo et al., 2019). In this work, we focus on deep learning classification using the structural connectomes derived from diffusion-weighed MRI.

In structural and functional connectivity analysis, the human brain complexity is represented as an interconnected network. This connectome is a graph whose nodes are brain anatomical regions and edges are connectivity "strength". Several studies explored brain networks using functional imaging modalities (Prescott et al., 2014; Filippi et al., 2018). The knowledge and characterization of this connectome, and underlying changes in brain structure and activity, is essential to study cognitive and behavioral impairments.

Both fMRI and dMRI connectivity matrices have been used widely for studying AD due to the rich information they held. Prescott et al. (2014) studied the differences in the structural connectomes among patients with normal cognition (NC), mild cognitive impairment (MCI), and AD while discovering associations between the structural connectome and cortical amyloid deposition. Changes in weighted structural connectome metrics were observed between NC, MCI and AD, with decreases from the NC group to the MCI and AD groups. Filippi et al. (2018) investigated the structural and functional brain connectomes in patients with AD and MCI. Severe graph analysis abnormalities were distinguished for both the functional and structural connectomes in AD patients compared to NC, where all brain lobes are involved except the basal ganglia and parietal lobes. Ye et al. (2019) observed connectome abnormalities between different phases of the AD. Results underlines 13 brain regions involved in the disease.

In this paper, we intend to explore to what extent a deep convolutional neural network trained on the connectome of a large number of ADNI subjects can help underline the characteristics of the AD structure (adni.loni.usc.edu). In that perspective, we trained a modified version of the BrainNetCNN (Kawahara et al., 2017) on the connectivity matrices of a heterogeneous set of patients to predict three groups of subjects: normal control (NC), mild cognitive impairment (MCI) and AD. Then, with the help of visualization techniques and a thorough node ablation analysis, we get to visualize brain regions as well as their connectivity that are involved in the prediction of AD.

Our work is a contribution to more explainable AI in advance medical imaging, using deep learning to better understand the specific connectivity of AD through connectivity ablation analysis and saliency map extraction, for understanding how the brain connectivity differs and change based on the different brain's alteration with dementia.

## 2. Methods

Since structural connectomes from DW-MRI tractography contain edge weights between pairs of regions, they can easily be represented by a 2D matrix of all connections (Jeurissen et al., 2019). This matrix is an adjacency matrix $A$ of size $N \times N$, where $N$ is the number of regions and $A_{i,j}$ is the weight between regions $i$ and $j$. While this connectivity matrix can be pictured as a 2D image (c.f. Figure 1 for an example of a connectivity matrix), it cannot be inputted directly into a regular convolution neural networks (CNN), as the local neighborhood around each element $(i, j)$ is not isotropic. This is due to the very nature of

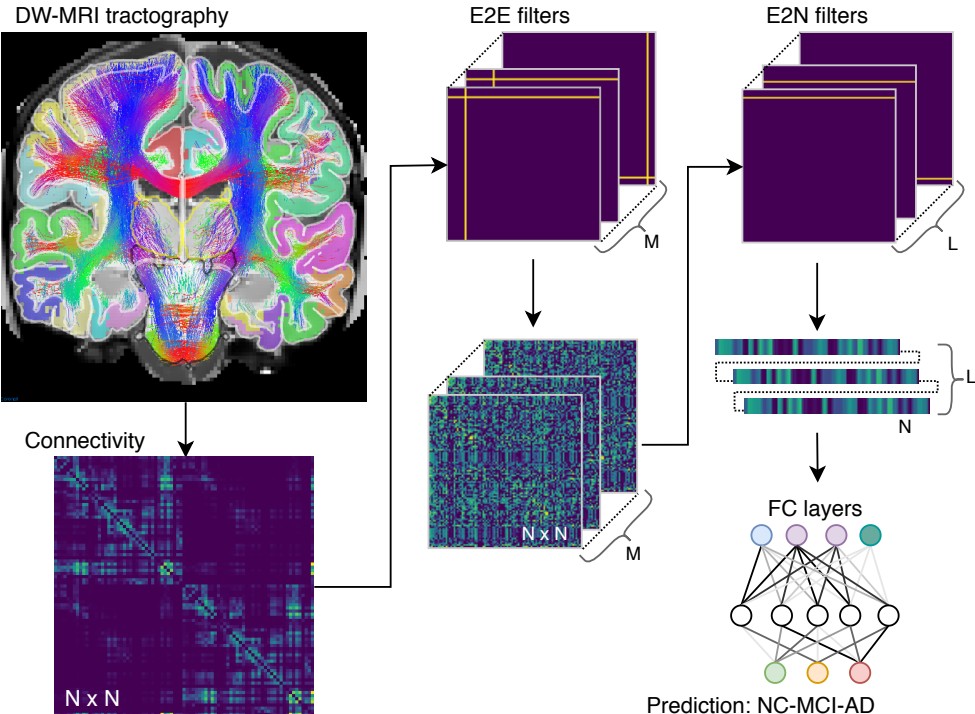

Figure 1: The BrainNetCNN-like architecture of our method.

a connectivity matrix, where neighbors of a node are distributed along horizontal lines and columns, i.e. connections to different brain regions.

As such, we implemented a modified version of BrainNetCNN (Kawahara et al., 2017) originally used to predict cognitive and motor activities in premature infants. This network implements two convolution filters adapted to the context of a connectivity matrix: a so-called *edge-to-edge* (E2E) filter and an *edge-to-node* (E2N) filter.

**Edge-to-Edge (E2E) and Edge-to-Node (E2N) layers**

By definition, each region $i$ of a symmetric adjacency matrix $A$ is connected to all elements in the $i^{\text{th}}$ row ($A_{i\cdot}$) and $i^{\text{th}}$ column ($A_{\cdot i}$). As such, it is not related to its local neighbors like a pixel of a regular image would be. Thus, if such adjacency matrix is to be processed by a CNN, the usual 2D convolution operation need to be redefined. In that perspective, the E2E filter is made of two 1D convolution filters: one spanning along the rows of $A$ and one spanning along the columns of $A$. These filters all process the connectivity matrix $A$ and produce M feature maps as shown in Figure 1. This simple, but effective operation, filters the topological locality of brain networks, combining weights of connected edges.

After the E2E layer comes the E2N layer. The E2N filters extract features from all the weights in each row in the preceding feature maps and convert it into a single scalar. The E2N layers captures the second order connectivity (indirect connections) of the brain. As shown in Figure 1, the E2N layer is formed of $L$ filters and thus returns $L$ vectors of dimensions $N \times 1$. We implement this operation with a 1D convolution filter.

**The proposed architecture**

Our proposed network architecture is formed of an E2E layer followed by an E2N layer and two fully-connected (FC) layers. The E2E and E2N layers are hierarchical brain networks feature extraction functions. Note that for better convergence during training, a batch-norm operation was added at the end of each layer. These layers are then followed by two FC layers and then the output layer. To obtain a binary classification (e.g. NC-AD), a *sigmoid* function is added to the output layer. Otherwise, for a three-class prediction (NC-MCI-AD), a *softmax* function is used at the output.

HpBandSter (2018) algorithm was used to fine-tune the model's hyperparameters including the number of E2E and E2N filters and the activation function (*ReLu* or *LeakyReLu*). This algorithm relies on Falkner et al. (2018) algorithm which combines both *HyperBand* (Li et al., 2017) and Bayesian optimization algorithm.

The final best architecture is formed of a E2E layer with 23 filters and a *LeakyReLu* activation function with a negative slope of 0.015976 followed by an E2N layer with 59 filters with a *ReLu* activation function. Meanwhile, the FC layers each contain 19 units, the learning rate is 0.007812 and the batch size is of 25. Subsequent good configurations among the other possible ones are reported in the appendix.

**Training**

The proposed architecture was implemented using *Pytorch* with the use of *HpBandSter*, to optimize all hyperparameters. In order to ensure that configurations with best performance were retained, 10 fold cross validation was utilized. The loss function is the well-known cross-entropy which we minimize with the Adam optimizer. For the learning procedure, along with the 10-fold cross-validation, the dataset was split into three sets: training (70%), validation (15%) and test (15%).

## 3. Dataset

We used the ADNI (Alzheimer's Disease Neuroimaging Initiative) dataset (Iwatsubo, 2011) which is a well-known a longitudinal dataset with different time acquisition for the images, baselines and after 6 months, 12 months, 24 months, 48 months, where baseline images are MRI images acquired for the first time (day 1). Different releases of ADNI exist. Here, ADNI2 and ADNI-Go were used with clinical-like DW-MRI acquisitions with 41 directions, 2mm isotropic and b-value 1000 s/mm$^2$. In order to compute the tractography from DW-MRI, Theaud et al. (2020) pipeline with default parameters was employed, ensuring reproducibility and fast processing. Connectivity matrices were estimated from tractography using the "Lausanne 2008" brain parcellation, an atlas of anatomical regions (Dale et al., 1999; Hagmann et al., 2008), and streamline count between every pair of regions (see Figure 1).

After quality control sanity checks, the final dataset is formed of 480 connectivity matrices distributed as follows: 152 NC, 181 MCI and 147 AD. Meanwhile, the baseline sample is formed of 57 NC, 95 MCI and 34 AD. The connectivity matrices were obtained with the Freesurfer Desikan-Killiany parcellation tool (Desikan et al., 2006). This resulted into 83 regions : 68 cortical regions, 14 subcortical (nuclei) regions and 1 brainstem region.

These connectivity matrices represent an undirected (symmetric) complete weighted graph of dimensions $83 \times 83$. It is worth mentioning that connectivity matrices were normalized to sum to 1, in such a way that each element of the matrix represents a probability of a tractography connection occurring between region $i$ and region $j$ in the brain. The Figure 1 (bottom-left) presents a connectivity matrix from DW-MRI tractography in between cortical regions.

Furthermore, it is known that streamline count between regions is heavily dependent on the size or surface area of cortical regions (Girard et al., 2014). Hence, the local volume of each cortical region was added in the previously generated connectivity matrices. In order to obtain the connectivity matrices with the cortical volume, the diagonal of the initial connectivity matrices has been filled with cortical volume of each region. This diagonal was also normalized, summing to 1, resulting in reconstructed connectivity matrices with a total sum equal to 2. As a result, our experiments were tested for both types of matrices, with and without cortical region volume in the diagonal.

## 4. Experiments

As mentioned before, we hypothesize that the use of a trained CNN can help better understand the specifics of the AD connectomics. We do this through two experiments: 1) an ablation analysis to measure to which extent a region and/or an edge can affect the prediction of the neural network and 2) a visualization experiment to recover which areas of the brain drive the most of the output of the neural network.

### Regions and connections ablation analysis

The main idea here is to change the connectivity between regions of the brain in order to evaluate the impact of that change on the overall performance of the model. For this, three approaches were implemented: a *node ablation*, a *node randomization* and an *edge ablation*. The *node ablation* forces to zero the connections between a region $i$ and every other regions. The *node randomization* "randomizes" values of connectivity between a region $i$ and the other regions while keeping the same average instead of forcing them to zero. As for *edge ablation*, we set to zero the connection between regions $i$ and $j$. This last approach is also tested with a combination of edges, to a maximum of 4 connections simultaneously. The ablation analysis is done in turn for each region and each edge.

### Saliency map extraction

The goal of the ablation analysis is to identify if a node or an edge is responsible on its own for certain predictions of the network. In this section, we want to identify if a group of nodes or edges is responsible for certain outputs of the neural net. Hence, a legitimate step to understand the regions driving the model prediction is by retroprogating the gradient from a maximally activated output neuron all the way to the input connectivity map $A$. We did so after training the network to discriminate between NC, MCI and AD. The magnitude of the gradient shall thus give us a clue on which combination of regions and edges are most important for predicting these classes.

For interpretability and explainability of the results, advice from an expert neuroanatomist are included in the discussion. Hence, the interpretation will depend on the resulting features visualized over averaged inputs for each class and prior knowledge from the AD literature.

## 5. Results

Results given in this section are from the learned model applied to the test set, after training and validation, including a 10 fold cross-validation. The accuracy of the one-to-one prediction are as follows: 78% for NC-MCI (45 test samples), 91% for NC-AD (45 test samples) and 81% MCI-AD (49 test samples). Table 1 summarizes the one-to-one predictions with the following metrics: prediction precision, recall, F1-score, accuracy of training, validation and test. For both datasets: with and without cortical volume for each region in the diagonal (Table 1). For the one-to-all prediction, NC-MCI-AD (72 test samples), when incorporating cortical region volume in the diagonal of the matrices, the score improves from 76% to 78%.

| Prediction | Cortical regions volume | precision | recall | F1-score | valid. acc. | test acc. |
|---|---|---|---|---|---|---|
| NC - MCI | | 86% | 70% | 77% | 79% | 78% |
| NC - AD | no | 95% | 86% | 90% | 85% | 91% |
| MCI - AD | | 78% | 81% | 80% | 71% | 81% |
| NC - MCI | | 74% | 74% | 74% | 77% | 72% |
| NC - AD | yes | 91% | 91% | 91% | 95% | 91% |
| MCI - AD | | 80% | 90% | 85% | 75% | 86% |

Table 1: Reported metrics for the experiments with and without regions volume.

**Regions and connections ablation analysis**

To our surprise, shutting down nodes and edges did not decrease in any significant way the predicted scores. In addition, the *node randomization* decreased our prediction accuracy to 50%, which emphasizes the importance of the structure within connectomes and how regions are connected between each other.

**Saliency map visualisation**

As mentioned before, we retroprapagated the gradient from the maximally activated output neuron associated with AD, MCI and NC. Regions with higher values in the AD and MCI saliency maps are: hippocampus, amygdala, parahippocampal, entorhinal, fusiform regions.

To further illustrate the difference between the activated regions of AD, MCI and those of NC, we subtracted the NC saliency map from the AD and the MCI. We did so with the purpose of underlying the specifics of the AD and MCI connectomes. The subtracted saliency maps are illustrated in Figure 2. This revealed that the entorhinal was the most intense difference between AD and NC along with hippocampus for MCI and NC. These

regions are reported in AD research from voxel-based morphometry, cortical thickness or functional connectomics (Jhoo et al., 2010; Choo et al., 2010; Pennanen et al., 2004; Hojjati et al., 2017).

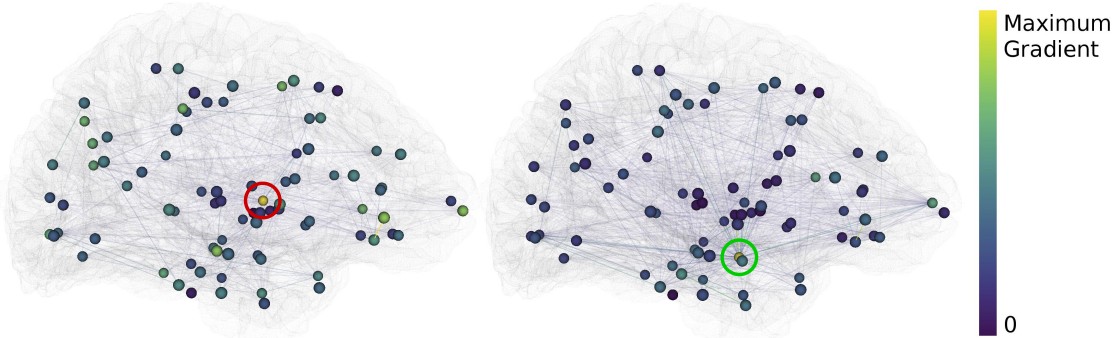

Figure 2: Saliency map features visualization resulting from the difference of two saliency maps generated with the one-to-all model: A) MCI minus NC, B) AD minus NC. Diagonal values are presented with node color, and other values represented with edge color. The red and green circle are around the hippocampus and entorhinal nodes, two regions known for their implication in AD.

## 6. Discussion

We have shown that CNNs adapted to the structure of DW-MRI tractography connectomes can be used to classify MCI and AD afflicted brains. Moreover, we showed that our trained network could be used to gain insights into the structural connections that drive the AD pathology.

**AD prediction**

Previously reported results for MCI and AD prediction are in the order of 80%, e.g. 60%-70% from MRI morphological methods (Lisowska et al., 2019), and from 80%-90% with functional MRI approaches (Hojjati et al., 2017). On the other hand, Abrol et al. (2020) proposed a deep residual neural network for predicting the progression of AD, achieving a median accuracy of 91% for AD vs NC, 86% for both MCI vs NC and MCI vs AD. While the method published in Li et al. (2015), consists of a deep learning neural network to identify the progression of AD based on MRI and PET modalities, while using advanced techniques for improving the model prediction like dropout, stability selection, adaptive learning factor and multitask learning strategy, and reported an accuracy of 91.4% for AD vs NC, 77.4% for MCI vs NC and 70.1% for MCI vs AD.

Our work shows competitive prediction percentages and also emphasizes that the key challenge in AD prediction is the prediction between MCI and AD, and between NC and MCI. These are the hardest classification tasks, where disease prediction is not clear-cut, and most likely requires more information (multi-modality, genetics, amongst others).

### Regions and connections ablation analysis

The results of the ablation procedure support the idea that **no** single region and its connections are responsible for AD prediction, but the combined effect of several cortical regions, that are directly or indirectly connected via long-range fiber tracks. By indirect connectivity, we mean that a 2nd order connectivity exists between these regions. DW-MRI tractography is the only non-invasive modality that can provide this structural connectivity brain architecture, which is essential and should be considered in future AD studies.

### Saliency map visualization

The amplitude of the retropropagated gradient underlines which regions strongly correlate with the neural net prediction. However, this correlation could be explained by a lower or higher structural connectivity estimated from the DW-MRI thus the analysis of the saliency map should be interpreted with care.

### Limitations and future directions

One of the current limitations of our work is the absence of anatomical priors for the structural connectome reconstruction. As such, more insights from the disease along with anatomical constraints could improve results. Since incorporating cortical region volumes can improve the prediction, adding more information from relevant brain features could, furthermore, increase the model power. For example, more information from diffusion such as fractional anisotropy (FA), mean diffusivity (MD), as well as more information from other MRI contrasts (e.g. cortical thickness, myelin, functional connectivity). As a result, future direction in predicting AD, and it's progression with MCI, is within the application of advance geometric or graph CNN over the connectome (Bronstein et al., 2017). Furthermore, along with continuous progress and efforts in creating larger datasets, a regression problem for AD progression prediction could be formulated, so that the disease progression can be assessed as a continuum in time.

## 7. Conclusion

In this paper, we conducted an explainable AI experiment to better understand the connectomic structure of the AD. From a CNN trained on the brain connectomes of ADNI patients, we showed from an ablation procedure that no single region is responsible for AD, but the combined effect of several cortical regions. We also showed that the entorhinal is the most intense difference between AD and NC along with hippocampus for MCI and NC. These regions are reported in AD research from voxel-based morphometry, cortical thickness or functional connectomics ((Jhoo et al., 2010); (Choo et al., 2010); (Pennanen et al., 2004); (Hojjati et al., 2017)). Our findings thus show that deep convolution networks can be used to gain insights into the specifics of a neurodegenerative disease such as AD. This could have important implications in neurodegenerative diseases analysis.

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
