# OpenReview forum: "Understanding Alzheimer disease’s structural connectivity through explainable AI"
_MIDL.io/2020/Conference — MIDL 2020_

### Official Review · AnonReviewer2 · 2020-03-02
**Lack of novelty**

**Rating:** 1
**Confidence:** 5
**Recommendation:** Poster

**Summary:**

This paper introduces a modification of the existing BrainNetCNN for AD/MCI diagnosis with DW-MRI. Unlike the original BrainNetCNN, the authors exploited the nature of the adjacency matrix that represents the connections among regions by defining two 1D convolution filters in E2E. Further, it is also considered to use the regional volume features in order to reflect the difference in size among regions when constructing a connectivity matrix. The ablation-based analysis was also conducted to verify the validity of the work.

**Strengths:**

It is of great interesting in the field to use deep learning methods for DW-MRI analysis.
The idea of taking the regional volume information into the model combined with an adjacency matrix is reasonable.

**Weaknesses:**

The technical novelty is minor and the descriptions on the proposed method is not clear.

No comparison with other method(s), at least with BrainNetCNN is required.

More rigorous analysis in ablation and saliency map extraction is expected.

**Detailed Comments:**

Please refer to "Questions To Address In The Rebuttal".

**Justification Of Rating:**


The technical novelty is minor and the description and experiments are insufficient.

No comparison with other method(s), at least with BrainNetCNN is required.

More rigorous analysis in ablation and saliency map extraction is expected.

**Paper Type:**

validation/application paper

**Questions To Address In The Rebuttal:**

The operation of E2E filtering with two 1D convolution filters is not clear. From the NxN matrix, how the M number of NxN matrices are obtained? It would be more clear with equations.

To this reviewer’s understanding, each of the rows (or columns)  in the adjacency matrix A are basically defined in different spaces, where each denotes the connectivity from the i-th region to other regions. In this regard, it doesn’t make sense to use the shared filters across rows or columns.

“Meanwhile, the baseline sample is formed of 57 NC, 95 MCI and 34 AD. These connectivity matrices represent an undirected (symmetric) complete weighted graph of dimension 83 × 83.”: What do the authors mean by “baseline” and where the graph of 83x83 in dimension come from?

To this reveiewer the “Regions and connections ablation analysis and saliency map extraction” are related to the so-called “sensitivity analysis” in explainable AI. Rather than setting zeros, it would be better to compute and analyze a Jacobian matrix combined with the inputs.

**Special Issue:**

no

---

> ### Author Response · Authors · 2020-03-27
> **Baseline samples & 83x83 dimensions.**
>
> Baseline samples are the connectome matrices built from the so-called “ADNI baseline MR images”. Since ADNI is a longitudinal dataset, baseline images are those that were acquired at day 1 of the acquisition process. As such, ADNI contains baseline images as well as MR images acquired after 6, 12, 24 and 48 months. This will be clarified in the revised version of the manuscript.
>
> As for the 83x83 dimension of the matrix, it comes from the Freesurfer Desikan-Killiany  parcellation, using 68 cortical regions, 14 subcortical (nuclei) regions and 1 brainstem region.
> Rahul S. Desikan et al. An automated labeling system for subdividing the human cerebral cortex on MRI scans into gyral based regions of interest, Neuroimage, 2006, 31 (3),
> 968--980,
> https://surfer.nmr.mgh.harvard.edu/fswiki/CorticalParcellation

---

> ### Author Response · Authors · 2020-03-27
> **No comparison with other methods like BrainNetCNN**
>
> We propose an explainable AI framework to better understand how AD affects brain connectivity.  As such, the contribution isn’t so much about how the proposed CNN can outperform previously published CNNs on the task of brain classification, but how a CNN whose structure is tailored to connectivity matrices can reveal information on AD-specific brain connectomic.  Since BrainNetCNN has not performed any explainable AI, it would be hard to compare our findings with theirs.  In fact, to our knowledge, no other paper has applied explainable AI on structural connectivity matrices (from diffusion MRI) to understand the specifics of a neurodegenerative disease.  Furthermore, our CNN architecture is very close to that of the BrainNetCNN and so, on the task of brain classification, would probably have similar results.  Again, our goal is not to propose a novel CNN architecture that would outperform previously published CNNs (like BN-CNN) but to employ CNN to explore and better understand brain connectivity.

---

> ### Author Response · Authors · 2020-03-27
> **E2E Filtering.**
>
> Concerning the operation of E2E filtering with two 1D conv filters, a 1D convolution is applied to rows, and the other is applied to columns, both applied to each region.  Each E2E filter for an input $A$=$(A^n)_{n \in [\![1, M]\!]}$ where $A^n$=$(A^n _{i, j}) _ { (i, j) \in [\![1, N]\!]^2}$, either an adjacency matrix with $M=1$ for the first E2E layer or a set of feature maps with $M$ is the number of generated feature maps, is mathematically expressed by $B$=$(B _{i, j}) _ { (i, j) \in [\![1, N]\!]^2}$. \\
> Where $B_{i, j} = \sum_{n=1}^{M} \sum_{k=1}^{N} A^{n} _{i, k} * r_{k}  + A^{n} _{k, j}*c_{k}$, with $r=(r_{k})_{k \in [\![1, N]\!]}$ and $c=(c_{k})_{k \in [\![1, N]\!]}$ are the learned weights for the first and second 1D convolution layer respectively.
>
> Hence, from the $NxN$ matrix, after applying an E2E layer formed of $M$ E2E filters, we obtain in the output $M$ matrices of size  NxN. In https://kawahara.ca/convolutional-neural-networks-for-adjacency-matrices/, a visualization of E2E filters and mathematical equations for the operation from the original BrainNetCNN.

---

### Official Review · AnonReviewer1 · 2020-03-11
**Good validation of existing method**

**Rating:** 3
**Confidence:** 5
**Recommendation:** Poster

**Summary:**

The authors build a CNN classifier for AD/MCI/CN on ADNI data based on DTI connectivity data. The method has been published before in another application. Several validation experiments are performed to assess the robustness of the network and important nodes/regions using saliency maps. Classification performance is good.

**Strengths:**

Method is very suited for this important application.
Validation experiments are well-designed: both the ablation analysis and the saliency analysis are useful.
The classification performance is placed into the context of the literature.

**Weaknesses:**

It is not completely clear to me what the added value of this work is.
No reference methods for classifcation performance are included. It would be valuable to know what the performance of for example a conventional classifier based on the same input data would be. Or, similarly the performance of a CNN classifier on more raw diffusion/tensor/tractography data.
New insights based on validatoin experiments are very limited.


**Justification Of Rating:**

Important application. Validation experiments are interesting and classification results are good. Unfortunatly, there is not much novelty in methodology or new insights resulting from the validation experiments.

**Paper Type:**

validation/application paper

**Questions To Address In The Rebuttal:**

Section 4: "Advice from an expertent in the field are included in the discussion". This is very strange to me? Aren't the authors experts in the field, what is meant by this comment?

**Special Issue:**

no

---

> ### Author Response · Authors · 2020-03-27
> **About Novelty**
>
> In this paper, we experimented how far we could go to better understand the brain connectivity of AD patients with the blind use of a CNN. As of today, no one has ever used DL as a means for doing so through regions and connections ablation analysis and saliency map extraction. Previous methods for understanding which connexions in the brain are associated with AD were never used as such (Prescott et al.,2014, Filippi et al., 2018 and Ye et al. 2019). In the paper, we show that the amplitude of the retro propagated gradient can be used to underline brain regions that differ between AD, MCI and NC.  Even more interesting, these regions have also been recognized in previous (none-deep-learning) works  [Jhoo et al., 2010; Choo et al., 2010; Pennanen et al., 2004; Hojjati et al., 2017]. Our paper should thus be regarded as an explainable AI paper rather than a paper proposing a new method for classifying brains.
>
> Prescott et al.,2014 : The Alzheimer  structural  connectome:  changes  in  cortical  network  topology  with  increased amyloid plaque burden. Radiology, 273(1):175–184, 2014
> Filippi et al., 2018 : Changes in functional and  structural  brain  connectome  along  the  Alzheimers  disease  continuum. Molecular psychiatry, page 1, 2018
> Ye et al. 2019 : Connectome-wide network analysis of white matter connectivity in Alzheimer’s disease. NeuroImage:  Clinical, 22:101690, 2019

---

> ### Author Response · Authors · 2020-03-27
> **Section 4: "Advice from an expert in the field are included in the discussion". This is very strange to me? Aren't the authors experts in the field, what is meant by this comment?**
>
> By expert, we mean a neuroanatomist.  None of us are neuroanatomists.

---

> ### Author Response · Authors · 2020-03-27
> **About to reference to other methods as comparisons**
>
> Due to space limitation, here are results from 2 other classification method which do not use connectivity matrices : Abrol et al. 2018  achieved a median accuracy of
> AD vs NC             MCI VS NC                   MCI vs. AD
>      91%,                      86%,                               86%
>
> while in Li et al 2015, the proposed CNN get the following accuracies
> AD vs NC          MCI vs NC                MCI vs AD
>    91.4%                 77.4%                       70.1
>
> Let us remind that we get
> AD vs NC          MCI vs NC                MCI vs AD
>      91%   	        78		      81
>
> which is similar to these other methods.
> Anees Abrol et al. : Deep Residual Learning for Neuroimaging: An application to Predict Progression to Alzheimer’s Disease, 2018 Bioarxiv, for the Alzheimer’s Disease Neuroimaging Initiative
> Feng Li et al.: A Robust Deep Model for Improved Classification of AD/MCI Patients.IEEE J Biomed Health Inform. 2015 Sep; 19(5): 1610–1616.

---

### Official Review · AnonReviewer4 · 2020-03-14
**Alzheimer’s disease classification using CNN over structural connectomes**

**Rating:** 2
**Confidence:** 3

**Summary:**

This paper uses deep neural networks for Alzheimer's disease classification using model trained on diffusion weighted MRI. Paper utilized structural connectivity matrices. Findings of papers support that connnectomics information from diffusion MRI tractrography is useful for understanding biomarkers of Alzheimer's disease.

**Strengths:**

This paper explores the idea of classifying Alzheimer’s data  by using CNN. This is a very interesting topic in DL-based medical image processing, therefore perfectly fits in the scope of the conference, because human radiologists also consider taking aid from other resources. Unfortunately, this paper is making only very limited steps towards achieving its ambitious goal.

**Weaknesses:**

People are using structural connectivity matrices since long in the literature. It has sort of become obsolete to have classifaciton done with this connectivity matrices. Where does the paper stands in terms of novelty is not clearly mentioned. Motivation to the problem is not clear from introduction. Perhaps, citations of more recent work utilizing structural connectivity matrices for Alzheimer's disease classification is necessary to motivate the readers.

**Justification Of Rating:**

The topic is very relevant and interesting. The only reason I'm not recommending acceptance is lack of comparison with the literature. More citations of recent work is required along with comparisons with the proposed approach.

**Paper Type:**

both

**Special Issue:**

no

---

> ### Author Response · Authors · 2020-03-27
> **Limited step towards this goal and comparisons with recent work**
>
> The overarching objective of this paper is to propose ways to use DL methods to better understand the specific connectivity of AD through connectivity ablation analysis and saliency map extraction.  To our knowledge, no work in the literature has done that before.  Our work can be seen as contributing to “explainable AI”. Said otherwise, this paper isn't so much about finding yet a new method to best classify AD vs MCI vs NC, but about understanding how the connectivity of those brains differs and how DL can help us with that.  For example, results of the ablation procedure support the idea that no single region is responsible for AD, but the combined effect of several cortical regions.  Also, as mentioned in page 6 “the entorhinal is the most intense difference between AD and NC along with hippocampus for MCI and NC. These regions are reported in AD research from voxel-based morphometry, cortical thickness or functional connectomics (Jhoo et al., 2010; Choo et al., 2010; Pennanen et al., 2004; Hojjati et al., 2017)”. We believe that these findings are not trivial and show that deep conv nets can be used to understand the specific of a neurodegenerative disease such as AD.

---

> ### Author Response · Authors · 2020-03-27
> **has sort of become obsolete to have classifaciton done with this connectivity matrices**
>
> We agree that fMRI and dMRI connectivity matrices have been used extensively in the AD, but we do not agree that they are obsolete. We are citing works that use dMRI connectivity matrices but we might have omitted some due to space restrictions. Moreover, it is important to note that the geometric structure of the dMRI connectivity matrices has been exploited here, in our current work, using the brainNetCNN method. Again, to the best of our knowledge, this is a first in the AD litterature.

---

### Author Response · Authors · 2020-04-02
**Feedbacks on the rebuttal**

Dear reviewers,

we would like to know if you had time to look at our answers to your comments. We would be happy to know what you think about it, if you have any further questions that we could answer and if our answers had an influence on your rating.

yours truly,

The authors

---

### Meta-Review · Area_Chair1 · 2020-04-06
**MetaReview of Paper304 by AreaChair1**

**Rating:** 3
**Recommendation For Accepted Papers:** Poster

**Metareview:**

While the novelty of the paper is limited, I do believe that the novel application of DL to DWI connectivity for classification could spark some interesting discussion at the conference.

**Paper Type:**

validation/application paper

**Special Issue:**

no

---

### Decision · Program_Chairs · 2020-04-11

Accept